# The Functions of Cholera Toxin Subunit B as a Modulator of Silica Nanoparticle Endocytosis

**DOI:** 10.3390/toxins15080482

**Published:** 2023-07-29

**Authors:** Eva Susnik, Sandor Balog, Patricia Taladriz-Blanco, Alke Petri-Fink, Barbara Rothen-Rutishauser

**Affiliations:** 1Adolphe Merkle Institute, University of Fribourg, 1700 Fribourg, Switzerland; eva.susnik@unifr.ch (E.S.); sandor.balog@unifr.ch (S.B.); alke.fink@unifr.ch (A.P.-F.); 2International Iberian Nanotechnology Laboratory, Water Quality Group, 4715-330 Braga, Portugal; patricia.taladriz@inl.int; 3Department of Chemistry, University of Fribourg, 1700 Fribourg, Switzerland

**Keywords:** cholera toxin subunit B, silica nanoparticles, cytokines, endocytosis, macrophages, intestinal epithelial cells

## Abstract

The gastrointestinal tract is the main target of orally ingested nanoparticles (NPs) and at the same time is exposed to noxious substances, such as bacterial components. We investigated the interaction of 59 nm silica (SiO_2_) NPs with differentiated Caco-2 intestinal epithelial cells in the presence of cholera toxin subunit B (CTxB) and compared the effects to J774A.1 macrophages. CTxB can affect cellular functions and modulate endocytosis via binding to the monosialoganglioside (GM1) receptor, expressed on both cell lines. After stimulating macrophages with CTxB, we observed notable changes in the membrane structure but not in Caco-2 cells, and no secretion of the pro-inflammatory cytokine tumor necrosis factor-α (TNF-α) was detected. Cells were then exposed to 59 nm SiO_2_ NPs and CtxB sequentially and simultaneously, resulting in a high NP uptake in J774A.1 cells, but no uptake in Caco-2 cells was detected. Flow cytometry analysis revealed that the exposure of J774A.1 cells to CTxB resulted in a significant reduction in the uptake of SiO_2_ NPs. In contrast, the uptake of NPs by highly selective Caco-2 cells remained unaffected following CTxB exposure. Based on colocalization studies, CTxB and NPs might enter cells via shared endocytic pathways, followed by their sorting into different intracellular compartments. Our findings provide new insights into CTxB’s function of modulating SiO_2_ NP uptake in phagocytic but not in differentiated intestine cells.

## 1. Introduction

Among the variety of NPs used in biomedical applications, amorphous silica (SiO_2_) nanoparticles (NPs) are beneficial as drug and gene delivery carriers [1,2], and as stabilizing agents in therapeutics [3]. Additionally, SiO_2_ NPs are used as food additives to improve taste, texture, and durability [4]. Humans are routinely exposed to SiO_2_ NPs on a daily basis through various routes, including oral ingestion [5]. The main target of orally ingested SiO_2_ NPs is the gastrointestinal tract [5], which is at the same time continuously exposed to noxious components, such as bacterial products [6]. In this study, we sought to understand the cell association with SiO_2_ NPs in the presence of a bacterial virulent factor, e.g., cholera toxin subunit B (CTxB). CTxB is a non-toxic subunit of the cholera toxin, secreted by the Gram-negative bacteria *Vibrio cholerae* [7]. It exhibits a strong binding affinity to gangliosides on the cell membrane, specifically to the monosialoganglioside (GM1), primarily concentrated in lipid-raft-rich membrane regions. GM1 can be found in various cell types, including gut epithelial cells and antigen-presenting cells [8]. Upon binding to GM1, CTxB is endocytosed via clathrin-/caveolin-dependent or clathrin-/caveolin-independent mechanisms [9] and enters a retrograde trafficking pathway through endosomes to the Golgi apparatus or endoplasmic reticulum [10]. In addition, CTxB is used as a vaccine adjuvant as it can elicit a set of signaling events related to cell proliferation and membrane ruffling (mitogen-activated protein kinases—MAPK) and cytokine production (nuclear factor kappa light chain enhancer of activated B cells—NFκB) [11,12]. However, there is still no consensus on whether cell exposure to CTxB elicits the secretion of pro-inflammatory or anti-inflammatory cytokines [12,13,14,15,16]. As both signaling pathways and lipid rafts play a crucial role in endocytosis [17], we hypothesized that treating cells with CTxB could potentially modulate the endocytosis and trafficking pathways of silica NPs. As a positive control, we treated cells with lipopolysaccharide (LPS), which is known to induce a pro-inflammatory response and promote the NP uptake in J774A.1 cells via binding to Toll-like receptor 4 (TLR4) [18]. Only a few studies have explored the use of CTxB to manipulate the endocytosis of NPs. As an example, Walker et al. functionalized silica NPs with CTxB and showed that the uptake of NPs into HeLa cells was facilitated by both clathrin- and caveolae-mediated mechanisms [19]. In this study, we investigated the morphological changes in J774A.1 macrophages and polarized Caco-2 intestinal epithelial cells upon stimulation with CTxB and LPS, as well as the secretion of pro-inflammatory cytokine tumor necrosis factor-α (TNF-α). Further, we evaluated the uptake of 59 nm SiO_2_-Bodipy fluorescein (BDP FL) NPs and explained possible mechanisms for a reduced NP uptake in macrophages in the presence of CTxB. Lastly, we investigated the colocalization and intracellular fate of both CTxB and NPs.

## 2. Results and Discussion

### 2.1. Surface Expression of GM1 and TLR4 Receptors

A sufficient expression of surface receptors is crucial for the binding of extracellular ligands. J774A.1 macrophages and Caco-2 intestinal epithelial cells were selected for this study. Macrophages represent one of the first cell types to interact with NPs and bacterial components in the gastrointestinal tract [20]. The non-phagocytic Caco-2 cell line is frequently used to mimic an intestinal epithelial barrier [21] and is one of a widely used cell line in human research in a diverse range of applications, including NP uptake [9,22]. To confirm the expression of the GM1 receptor, a high-affinity binding site for CTxB, we performed immunostaining experiments using antibodies against GM1, followed by visualization via confocal microscopy. Caco-2 cells were cultured on inserts for a period of 21 days to establish a polarized cell monolayer with apical and basal sides, as confirmed via tight-junctions staining (Appendix A) and transepithelial resistance measurements (TEER) (Appendix A). J774A.1 macrophages, on the other hand, were used 24 h after seeding on cell culture plates. The expression of the GM1 receptor was initially confirmed via confocal laser scanning microscopy on the plasma membrane of both cell lines (Figure 1a). However, our findings revealed a non-uniform distribution of GM1 expression specifically on the apical site of Caco-2 cell monolayers. This observation aligns with the study conducted by Jahn et al. [23], demonstrating that Caco-2 cells exhibit distinct morphological islands, wherein GM1 expression is detected solely in the undifferentiated cells within these islands. Additionally, we included the Western blot data, showing the GM1 expression (M_w_: 40 kDa) in both cell lines (Figure 1b). The GM1/GAPDH densitometry results showed no significant differences between Caco-2 cells and J774A.1 macrophages (Figure 1c).

In addition, we tested the expression of the TLR4 receptor, which is involved in the recognition and binding of LPS. Western blot analysis (Figure 1b) demonstrated the expression of TLR4 (M_w_: 95 kDa) in both cell lines. The TLR4/GAPDH densitometry results showed no significant differences between Caco-2 cells and J774A.1 macrophages (Figure 1c). Full Western blot images including the corresponding controls and weight marker are shown in Appendix A. The outcomes of these studies affirmed the suitability of using Caco-2 and J774A.1 cells for subsequent experiments.

### 2.2. The Cellular Morphology of Macrophages Is Altered by CTxB Treatment

The interaction of CTxB with the Caco-2 epithelial cell monolayer and J774A.1 macrophages was evaluated via confocal microscopy. Cell exposure to µg/mL CTxB resulted in the uptake of significant amounts of CTxB by J774A.1 macrophages (Figure 2a), whereas Caco-2 cells were able to internalize CTxB only to a limited extent, with CTxB uptake observed only in individual cells (Figure 2b). Several factors could account for the observed variations in cellular behavior: (i) the divergence in endocytic mechanisms between macrophages and epithelial cells [24], coupled with (ii) dissimilarities in their membrane structures [9] and (iii) the variable expression pattern of GM1 between cells [25]. In J774A.1 macrophages, exposure to CTxB resulted in the remodeling of the plasma membrane. Two independent studies have also reported the formation of similar membrane structures and proposed that cross-linking multiple glycosphingolipids is necessary for the CTxB-induced membrane deformation. This process leads to the generation of membrane curvature [26,27]. Although the precise molecular mechanisms underlying the CTxB-mediated membrane remodeling remain unclear, this mechanism represents a possible explanation. In Caco-2 cells, fainter and blurred F-actin staining was observed, but the cell borders were still visible. This can be due to an interference of the fluorescently labelled CTxB with the F-actin label. 

Additionally, cells were exposed to 1 µg/mL LPS to determine LPS-associated morphological changes. While Caco-2 cells showed no visible changes upon LPS stimulation, we observed the formation of cytoskeletal protrusions in J774A.1 macrophages, which may be attributed to the rearrangements of F-actin filaments, such as lamellipodia and filopodia [18,28].

The selection of LPS dose (1 µg/mL) and incubation time (24 h) was determined based on our previously published data, wherein we demonstrated the J774A.1 macrophage membrane remodeling and inflammatory response to the LPS [18]. In vitro studies on Caco-2 cells demonstrated that 1 µg/mL LPS modulates cells’ morphology and instigates the inflammatory response without causing cell death. For the choice of CTxB experimental conditions, we relied on the published findings of Schnitzer et al. [12] and Phongsisay et al. [13]. These studies provided valuable insights into the optimal conditions for studying the effects of CtxB on cellular responses.

### 2.3. Nanoparticle Characterization

To investigate the interaction between cells and NPs in the presence of bacterial components, we synthesized and characterized silica (SiO_2_) NPs that were fluorescently labeled with Bodipy fluorescein (BDP FL) dye. The core size of SiO_2_-BDP FL NPs was determined using TEM, and the hydrodynamic diameter of the NPs in Milli-Q water, cRPMI, and cMEM was determined using dynamic light scattering (DLS). Figure 3a,b represents a TEM micrograph and the core diameter distribution of 59 ± 6 nm SiO_2_-BDP FL NPs, respectively. The average hydrodynamic diameters of 20 µg/mL SiO_2_-BDP FL NPs were 85 ± 1 nm, 95 ± 2 nm, and 94 ± 2 nm measured at 24 h incubation at 37 °C in Milli-Q water, cRPMI, and cMEM, respectively. The correlation functions of 59 nm SiO_2_-BDP FL NPs obtained via DLS measurements in Milli-Q water and cRPMI at 24 h indicate no particle aggregation (Figure 3c). The NP polydispersity index (PDI) measured in Milli-Q water was 0.13, indicating a homogenous NP distribution [29] with a ζ-potential of—38 ± 1 mV. To investigate the influence of CTxB or LPS on NPs hydrodynamic diameter, we prepared three different NP suspensions: NPs alone, NPs mixed with 1 µg/mL CTxB, and NPs mixed with 1 µg/mL LPS. The average hydrodynamic diameters of SiO_2_-BDP FL, measured at 24 h of incubation at 37 °C in Milli-Q water, cRPMI, and cMEM in the absence and presence of CTxB and LPS via DLS, are shown in Table 1. The NPs’ hydrodynamic diameter remains stable in Milli-Q water, cRPMI, and cMEM in the presence of bacterial components. By using fluorimetry, we demonstrated that the NPs remained stable, and no detectable dye leaching was observed (Appendix A).

### 2.4. The Effect of CTxB on the Secretion of Pro-Inflammatory Cytokine TNF-α

CTxB can modulate the inflammasome complex in different cells by enhancing or suppressing the secretion of pro- and anti-inflammatory cytokines [11,12,14,32]. In this study, cells were exposed to 59 nm SiO_2_-BDP FL NPs alone or in combination with CTxB, either sequentially or simultaneously (as described in the Section 4). The secretion of TNF-α released in the cell culture medium of J774A.1 macrophages and the basal compartment of Caco-2 cells was quantified using enzyme-linked immunosorbent assay (ELISA). The results demonstrated that CTxB had no impact on the release of TNF-α in both cell lines (Figure 4a,b). Furthermore, the combined treatment of cells with CTxB and NPs did not exhibit a significant impact on TNF-α release. Our observation is consistent with the research conducted by Bandyopadhaya et al., where it was demonstrated that Caco-2 cells do not express discernible levels of TNF-α mRNA following exposure to *Vibrio cholerae* [33]. As anticipated, the stimulation of J774A.1 macrophages with LPS significantly increased in TNF-α expression across all exposure conditions [18]. Conversely, LPS did not have an impact on TNF-α production in Caco-2 cells. The lactate dehydrogenase (LDH) assay indicated the absence of membrane rupture following the administration of CTxB or NPs, suggesting no acute toxic effects. However, when J774A.1 macrophages were co-exposed to NPs and LPS, an elevated occurrence of membrane rupture was observed in J774A.1 macrophages (Appendix A), whereas no such effect was noted in Caco-2 cells (Appendix A).

### 2.5. The Effect of CTxB on the Cellular Association and Uptake of Silica NPs

To examine the potential influence of CTxB on the endocytic processes of J774A.1 macrophages and Caco-2 cells, as well as its subsequent effect on NP uptake, we employed confocal microscopy and flow cytometry. J774A.1 macrophages, known for their role as key phagocytes in the immune system, exhibited substantial internalization of NPs within 24 h (Figure 5a). In J774A.1 macrophages, we observed colocalization between CTxB and NPs during simultaneous co-exposure (PCC: 0.63) (Figure 5b) as well as sequential co-exposure with NPs (PCC: 0.82) (Appendix A), indicating a possibility for their uptake through shared endocytic mechanisms. The visualization of NP distribution in the cells in the presence of LPS under simultaneous and sequential co-exposure is shown in Figure 5c and Appendix A. In contrast, polarized Caco-2 monolayers did not display any internalization of NPs (Figure 5d), regardless of the presence or absence of bacterial components during simultaneous co-exposure (Figure 5e,f) or sequential co-exposure with NPs (Appendix A). These findings align with the observations made by Ye et al., who reported a restricted uptake of silica NPs in polarized Caco-2 cells [22]. Furthermore, a study performed by Ude et al. revealed that the cellular uptake of CuO NPs in differentiated Caco-2 cells accounted for less than 3% of the initial exposure dose [34]. This limited uptake can be attributed to the intestinal epithelium’s function as one of the initial barriers that impedes NP entry into organisms.

The semi-quantification of the intracellular SiO_2_-BDP FL NPs was conducted via flow cytometry analysis. The simultaneous exposure of J774A.1 macrophages to CTxB and NPs led to a ~two-fold decrease in NP uptake compared to single NP exposure (Figure 5g). In contrast, no uptake of NPs was observed in Caco-2 cells across all exposure conditions (Figure 5h). Based on these findings, we propose that the binding of CTxB to the GM1 receptor interferes with membrane components, including lipid rafts [35,36], and subsequently affects endocytosis-related pathways. This modulation ultimately leads to a reduction in NP uptake in macrophages. An alternative explanation for the reduced NP uptake observed following CTxB treatment could be that NPs are excluded from CTxB-positive endocytic vesicles. Consequently, the remaining endocytic pathways, including those involving NPs, may be downregulated due to using of plasma membrane resources for CTxB-positive vesicles.

J774A.1 macrophage exposure to LPS resulted in a ~two-fold increase in NP uptake, as previously shown [18].

### 2.6. Intracellular Localization of NPs in Macrophages in the Presence of CTxB

In the previous section, we described colocalization between CTxB and NPs during both simultaneous and sequential co-exposure, suggesting a possibility for their uptake through shared entry mechanisms, such as the clathrin-/caveolin-dependent or clathrin-/caveolin-independent pathways [9,37]. In this section, our objective is to investigate whether CTxB and NPs are localized within the same intracellular compartments. Upon deposition at the cell membrane, the NPs are internalized via endocytic mechanisms, then pass through early/late endosomes and accumulate in the lysosomes [38,39]. However, in contrast to NPs, CTxB can bypass lysosomes and accumulate in the endoplasmic reticulum or Golgi apparatus [19]. With this in mind, a colocalization of SiO_2_-BDP FL NPs and CTxB with early endosomes and lysosomes was investigated at 24 h upon exposure to J774A.1 macrophages. Following CTxB internalization, the pathways coalesced within early endosomes as evidenced by a Pearson’s correlation coefficient of 0.68 (Figure 6a). Consistent with the previous publications [40], CTxB was able to partially bypass lysosomes, as evidenced by the lower colocalization between CTxB and lysosomal staining (PCC: 0.43) (Figure 6b). In contrast, approximately half of the NP fraction already left early endosomes after 24 h of incubation (PCC: 0.50) (Figure 6c) and ended up in lysosomes (PCC: 0.63) (Figure 6d). This observation correlates well with our previous publications, showing a significant accumulation of 59 nm SiO_2_-BDP FL NPs in macrophage lysosomes at 24 h after exposure [18,41]. The divergent intracellular localization of CTxB and NPs can be ascribed to differentiate mechanisms of their endosomal/lysosomal transport. Moreover, the dissimilarities in the extracellular and intracellular mobility of CTxB and NPs must also be considered. Despite this, colocalization between NPs and CTxB observed under both simultaneous and sequential co-exposure conditions led us to conclude that a subset of NPs may still be present in the same endocytic vesicles as CTxB. We postulate that a subset of NPs enters cells alongside CTxB and is then transported into early endosomes or recycled to the cell membrane together with CTxB.

## 3. Conclusions

Upon ingestion, NPs can be internalized by immune and intestinal cells along with other components, such as bacterial toxins. In this study, we investigated the role of CTxB, which specifically binds to the GM1 receptor expressed on J774A.1 macrophages and differentiated Caco-2 intestinal cells, in modulating the uptake of 59 nm SiO_2_-BDP FL NPs. CTxB induced macrophage membrane remodeling, possibly through cross-linking glycosphingolipids and interference with lipid rafts. Interestingly, CTxB did not trigger TNF-α cytokine secretion. Our findings demonstrate that simultaneous exposure to CTxB reduced 59 nm SiO_2_-BDP FL NP uptake specifically in J774A.1 macrophages. On the other hand, no NP uptake was observed in polarized Caco-2 cells, which serve as a highly selective intestinal barrier. Although CTxB and NPs have distinct intracellular fate, we observed some degree of colocalization in macrophages. This colocalization suggests the potential uptake of CTxB and NPs through shared endocytic pathways. CTxB’s ability to reduce NP uptake in macrophages makes this molecule interesting for combination therapies with nanotherapeutics to avoid NP accumulation and degradation through phagocytic cells. As a further step, we propose exploring the mechanisms by which CTxB reduces NP accumulation in macrophages.

## 4. Materials and Methods

### 4.1. Nanoparticle Synthesis and Characterization

Fluorescently labeled SiO_2_-Bodipy fluorescein (SiO_2_-BDP FL) NPs of 59 nm diameter core size were synthesized in-house by following the Stöber method [42]. According to the previously described protocol [18], a mixture of 6.75 mL of Milli-Q water, 104 mL of absolute ethanol (VWR, Dietikon, Switzerland), and 3.9 mL of 25% ammonium hydroxide (Merck, Zug, Switzerland) was heated at 60 °C in a 500 mL rounded-bottom flask equipped with a reflux system. After 30 min, 11 mL of tetraethyl orthosilicate (TEOS) was added using a plastic syringe. Subsequently, a mixture of 200 µL of 10 mg/mL BODIPY FL N-Hydroxysuccinimide (NHS) ester in dimethyl sulfoxide (DMSO; Lumiprobe, Hunt Valley, MD, USA) and 4 µL of (3-aminopropyl) triethoxysilane (APTES) were added to the flask. The reaction was heated for an additional 4 h, cooled to room temperature, and dialyzed against Milli-Q water for five days using a 0.2 µm cellulose acetate filter. The NPs were stored in the dark at 4 °C. 

Imaging was performed using a Tecnai Spirit transmission electron microscope (TEM) (FEI, Hillsboro, OR, USA) operating at 120 kV equipped with a CCD camera (Eagle, Thermo Fischer, Waltham, MA, USA). The NPs core diameter and size distribution were calculated using ImageJ software (National Institutes of Health, Bethesda, MD, USA). To measure the hydrodynamic diameter, NP dispersions were characterized at 20 µg/mL in Milli-Q water and supplemented cell culture media (cRPMI and cMEM) at 37 °C using a dynamic light scattering spectrometer (LS Instruments AG, Fribourg, Switzerland) at the scattering angle of 90° and laser wavelength 660 nm. The NPs’ hydrodynamic diameter was also measured in the presence of LPS and CTxB at 1 µg/mL. The diameter of the NPs was determined by analyzing the obtained correlation functions using a previously described method [30,31]. Further characterization of the NPs’ polydispersity and ζ-potential was performed using phase-amplitude light scattering (ZetaPALS) in Milli-Q water (Brookhaven 90Plus Particle Size Analyzer, Brookhaven Instruments Corp., Holtsville, NY, USA). The NP concentration was assessed by measuring dry weights of NP suspensions after water evaporation at 70 °C.

### 4.2. Dye Leaching from SiO_2_ NPs

The potential release of dye from SiO_2_-BDP FL NPs after dialysis was assessed via fluorimetry. The NPs were subjected to high-speed centrifugation (16,000× *g*) for 1 h, and the supernatants were collected. To ensure a minimal number of NPs remained in suspension, the supernatants were subsequently centrifuged again at the same speed. As a control, NPs in Milli-Q water at the administered dose (20 µg/mL) were included in the experiments. Fluorescence emission intensity measurements were performed using a Fluorolog TCSPC spectrofluorometer (Horiba, Northampton, UK) equipped with the FluorEssence software (v3.8). For each sample, an emission spectrum ranging from 500 to 600 nm with a fixed excitation wavelength (λ_ex_) of 488 nm was recorded. The excitation and emission slits were both set to a fixed width of 4 nm, ensuring consistent measurement conditions across all samples.

### 4.3. Cell Culture

The mouse macrophage-like cell line J774A.1 (ATCC^®^, TIB-67^TM^, Rockville, MD, USA) was grown in Roswell Park Memorial Institute 1640 (RPMI) medium supplemented with 10% FBS (*v*/*v*), 2 mM L-Glutamine (100 Units/mL), and 100 µg/mL Penicillin-Streptomycin. At 80–90% confluence, cells were detached by scraping. Then, 0.52 × 10^5^ cells/cm^2^ were seeded in a 6-well plate (Corning, Reinach, Switzerland), with a growth area of 9.6 cm^2^ per well and a medium volume of 3 mL. For confocal microscopy imaging, 0.52 × 10^5^ cells were seeded in 8-well iBidi µ-Slide chambers (Cat. No. 80827, ibidi, Graefelfing, Germany), with a growth area of 1 cm^2^ per well and a cell suspension volume of 300 µL. Culturing was performed for 24 h at 37 °C in a humidified atmosphere with 5% CO_2_.

Caco-2 epithelial cells (ATCC^®^, HTB-37^TM^, Rockville, MD, USA) were cultured in complete Minimum Essential Medium (cMEM) supplemented with 20% FBS, 5 mL MEM non-essential amino acids (100 X), 2 mM L-Glutamine (100 Units/mL), and 100 µg/mL Penicillin-Streptomycin. Caco-2 cells were detached from the flasks at 80–90% confluence using a mixture of 0.05% Trypsin-Ethylenediaminetetraacetic acid (EDTA) (Gibco, Luzern, Switzerland). For each insert, 2 × 10^5^ cells in 0.5 mL cMEM were seeded onto the apical chamber of transparent cell culture inserts (Corning, Cat. #353103), with a growth area of 0.9 cm^2^, 1.0 µm pore diameter, and polyethylene terephthalate (PET) membranes for 12-well plates, with 1.5 mL cMEM applied in the basolateral chamber. After 2–3 days in the cell incubator, the medium in both chambers was replaced with fresh cMEM to remove non-adherent cells and avoid the formation of multiple layers. The cells were grown at 37 °C in a humidified atmosphere with 5% CO_2_ until day 21, when the brush border membrane and tight junction expression were fully developed.

### 4.4. Western Blot

Cells were lysed in lysis buffer (M-PER™ Tissue Protein Extraction Reagent, Cat. #78501, Thermo Fisher Scientific, Zug, Switzerland) supplemented with Halt^TM^ Protease Inhibitor Cocktail, EDTA-free (Cat. #78425, Thermo Fisher Scientific, Zug, Switzerland), and sodium fluoride (Cat. #27860, 20 mM, VWR, Dietikon, Switzerland). Plates were kept at 4 °C for 10 min and lysates were centrifuged at 14.000× *g* for 5 min. The protein content in the supernatant was quantified via a Nanodrop spectrophotometer (Thermo Fisher Scientific, Zug, Switzerland). The samples were boiled in a reducing Laemmli buffer for 5 min. Proteins (20 µg/lane) were electrophoretically separated on 12% polyacrylamide gels (Bio-Rad, Hercules, CA, USA) and transferred to polyvinylidene difluoride (PVDF) membranes at 150 mA for 75 min. A molecular weight marker mPAGE^®^ Color Protein Standard (Cat. #MPSTD4, Sigma-Aldrich, Buchs, Switzerland) was used to identify the corresponding detected bands. To check the efficiency of the transfer after staining, the membrane was soaked in 0.1% Ponceau-S (*w*/*v*) (Cat. #141194, Sigma-Aldrich, Buchs, Switzerland) and 0.05% acetic acid (*v*/*v*) in Milli-Q water for 10 min. Non-specific binding sites were blocked by incubating the membrane in 5% bovine serum albumin (BSA) in PBS (*w*/*v*) and 0.1% Tween 20 in PBS (*v*/*v*) (Cat. #P9416, Sigma-Aldrich, Buchs, Switzerland) for 1 h. The membrane was incubated with primary antibodies: anti-TLR4 (0.2 µg/mL, sc-293072, Santa Cruz Biotechnology, Heidelberg, Germany), anti-GM1 (1:1000, ab23943, Abcam, Cambridge, UK), and anti-GAPDH (1 µg/mL, sc-47724, Santa Cruz Biotechnology, Heidelberg, Germany), diluted in the blocking solution. The blots were incubated for 1 h at room temperature with goat anti-mouse HRP conjugated secondary antibody (Cat. #HAF007, R&D, Abingdon, UK) at 1:1000 (TLR4), goat anti-rabbit HRP conjugated secondary antibody (Cat. #HAF008, R&D, Abingdon, UK) at 1:1000 (GM1), and 1:5000 (GAPDH). Protein bands were visualized using the chemiluminescent HRP detection reagent Immobilon Forte Western HRP substrate (Cat. #WBLUF0020, Sigma-Aldrich, Buchs, Switzerland) under a chemiluminescence detection system (ImageQuant^TM^ LAS4000, Chicago, IL, USA). Washing with TBST (0.1% Tween 20 in TBS; *v*/*v*) was performed between each step. The average expression values of the indicated proteins were determined via densitometry (ImageJ software) from three independent experiments and normalized to the GAPDH control.

### 4.5. Cell Permeability

Caco-2 cell barrier integrity was assessed via transepithelial electrical resistance (TEER) measurement at days 2, 7, 14, and 21 after cell seeding. Cells grown on inserts were washed three times with PBS, and TEER was measured using disinfected (10 min in 70% ethanol) chop-stick Millipore equipment (Millicell ERS-2, EMD Millipore Corporation, Burlington, MA, USA). The TEER measurements were performed on three different spots of each membrane insert. Resistance values of two blank control inserts (medium without cells) were measured and subtracted from the values for cell layer samples. The absolute values were multiplied by the growth area of the membrane inserts (0.9 cm^2^) and represented as Ω × cm^2^.

### 4.6. Cell Exposures

J774A.1 cells were grown in 6-well plates or iBidi µ-Slide chambers for 24 h; 59 nm SiO_2_-BDP FL NPs were dispersed in 3 mL of complete RPMI medium (cRPMI), or 312 µL of cRPMI for cells cultured in iBidi chambers, at a final concentration of 20 µg/mL. Subsequently, the pre-mixed NP suspension [43] was applied to the cells and incubated for a duration of 24 h. Caco-2 cells, grown on inserts for 21 days, were exposed to 20 µg/mL of NPs in 0.5 mL cMEM applied on the apical side of the cell monolayer for 24 h.

The following experimental conditions were used: Single: The cells were exposed to 59 nm SiO_2_-BDP FL NPs dispersed in cRPMI/cMEM for 24 h.Sequential: The cells were initially pre-treated with either 1 µg/mL CTxB (C34778, Thermo Fisher Scientific) or 1 µg/mL LPS (Escherichia coli strain O111:B4, Cat. No. L4391, Sigma-Aldrich) in complete RPMI medium (cRPMI) or complete MEM (cMEM) for a duration of 24h. After the pre-treatment period, the CTxB or LPS was removed, and the cells were subsequently exposed to NPs dispersed in cRPMI or cMEM for an additional 24 h.Simultaneous: A combination of NPs (20 µg/mL) and CTxB (1 µg/mL) or LPS (1 µg/mL) was pre-mixed in cRPMI or cMEM and applied to the cells for 24 h.

After exposures, cells were washed 3 times with PBS to remove the non-cell-adhered NPs. All experiments were performed at 37 °C, 5% pCO_2_, and 95% relative humidity.

### 4.7. Cell Viability

Levels of the enzyme lactate dehydrogenase (LDH) in the supernatants were measured in triplicates according to the manufacturer’s protocols (Roche Applied Science, Mannheim, Germany). To evaluate cytotoxicity in Caco-2 cells, LDH activity was measured in the basal cell culture medium of the inserts. Application of 0.2% Triton X-100, diluted in the cell culture medium (*v*/*v*) for 24 h served as a positive control for membrane rupture. The absorbance of the colorimetric product formazan was determined spectrophotometrically (Benchmark microplate reader, BioRad, Cressier, Switzerland) at 490 nm, with a reference wavelength of 630 nm. LDH values are presented as a fold increase relative to untreated cells.

### 4.8. Enzyme-Linked Immunosorbent Assay (ELISA)

The amount of pro-inflammatory cytokine TNF-α in the cell culture media was assessed using enzyme-linked immunosorbent assays (DuoSet ELISA Development Kit) following the supplier’s protocol (R&D Systems, Zug, Switzerland). Secretion of TNF-α in the cell culture medium of Caco-2 cells was assessed from the basal cell culture medium of the inserts. A total of 100 µL of supernatant from each exposure was transferred to polystyrene high-binding surface 96-well plates (Corning, Switzerland) without dilution. Cells without treatments served as a negative control. Standards and samples were run in triplicates. The concentration of TNF-α was calculated based on the standard curves and fitted with a four-parameter logistic (4PL) approach using GraphPad Prism 8 software (GraphPad Software Inc., San Diego, CA, USA).

### 4.9. Immunofluorescence and Confocal Microscopy 

Cells were fixed in 4% paraformaldehyde (PFA, Cat. #158127, Sigma-Aldrich, Buchs, Switzerland) diluted in PBS (*v*/*v*) for 15 min. The membrane with Caco-2 cells were cut out from the inserts using a scalpel and transferred into new wells. Cells were permeabilized with 0.1% Triton X-100 (in PBS, *v*/*v*) for 15 min. 

Depending on the research question, different staining procedures were applied. To investigate the expression of the GM1 receptor, samples were immersed in primary antibody anti-GM1 (1:100, ab23943, Abcam, Cambridge, UK) for 1 h. After, the cells were incubated for 1 h at room temperature with goat anti-rabbit Alexa 555-conjugated secondary antibody (1:500, Cat. #ab150078, Abcam, UK). Tight junctions of Caco-2 cells cultured for 21 days were stained for zona occludens-1 (Zo-1) protein. The membranes were blocked with 1% BSA in PBS (*v*/*v*) and subsequently incubated with primary antibody anti-Zo-1 (1:100, Cat. #33-9100, Invitrogen, Thermo Fisher Scientific Inc., Waltham, MA, USA) overnight at 4 °C. Then, the membranes were incubated with goat anti-mouse IgG secondary antibody conjugated with Alexa Fluor 488 (1:1000, Cat. #ab150113, Abcam, UK) for 1 h at room temperature. Upon completion of the cell exposures to NPs, CTxB, and LPS, F-actin was stained with 0.66 µM Rhodamine-Phalloidin probe (Cat. #R415, Invitrogen, Thermo Fisher Scientific Inc., Zug, Switzerland) for 1 h. Cell nuclei were counterstained using DAPI (1 µg/mL in PBS, Cat. #D9542, Sigma-Aldrich, Buchs, Switzerland) for 10 min. All antibodies were diluted in 1% BSA (Cat. #A7030, Sigma-Aldrich, Zug, Switzerland) in PBS (*w*/*v*).

Caco-2 cells on inserts were mounted on a glass slide using Fluoromount aqueous mounting medium (F4680, Sigma Aldrich, Buchs, Switzerland), while J774A.1 cells were kept in 300 µL of PBS at 4 °C until imaging. The samples were washed three times with PBS between each step. 

Visualization of cells was conducted with an inverted Zeiss LSM 710 Meta microscopy (Axio Observer.Z1, Zeiss, Oberkochen, Germany) using an excitation laser of 405 nm (DAPI), 488 nm (Zo-1 and SiO_2_-BDP FL NPs), 561 nm (Rhodamine-Phalloidin and GM1), and 647 nm (CTxB) equipped with objective lens 63x(Plan-Apochromat 63x/1.4 Oil M27) or 40x (EC Plan-Neofluar 40x/1.30 Oil DIC M27) (Zeiss GmbH, Oberkochen, Germany). Representative images were collected and were further processed using the ImageJ-based software Fiji (National Institutes of Health, Bethesda, MD, USA).

### 4.10. Colocalization Analysis

The colocalization of SiO_2_-BDP FL NPs with CTxB was investigated by studying their simultaneous and sequential co-exposure. Furthermore, the colocalization of CTxB and NPs with early endosomes and lysosomes was assessed after a 24 h exposure. After exposures, cells were washed three times with PBS and incubated with fresh cRPMI supplemented with 75 nM LysoTracker Red DND-99 (Cat. #L7528, Invitrogen, Thermo Fisher Scientific, Zug, Switzerland) for 1 h to stain the lysosomes. Then, the LysoTracker was removed and cells were imaged live. For early endosome labeling, immunostaining with early endosome antigen 1 (EEA1) was performed. Cells were fixed and then immersed in 20 µg/mL of EEA1 (Cat. #ab109110, Abcam, Cambridge, UK) for 1 h. A secondary antibody goat anti-rabbit Alexa Fluor 555 (2 µg/mL, Cat. #ab150078, Abcam, Cambridge, UK) was added for 1 h. Cells were kept in PBS at 4 °C until imaging. All antibodies were diluted in 1% BSA in PBS (*w*/*v*). Imaging was conducted with an inverted Zeiss LSM 710 Meta microscopy using an excitation laser of 561 nm (LysoTracker Red and EEA1) equipped with objective lens of 40x or 63x.

The colocalization was calculated using ImageJ with the JACoP plugin and a custom-written script and represented as Pearson’s correlation coefficient (PCC) following the previously described protocol [41]. The magnitude of the PCC is a measure of the predictability of the relationship. Values close to +1 or −1 indicate a near-perfect ability to infer a Channel 1 intensity, given the corresponding Channel 2 intensity and vice versa [44]. Values near zero indicate that there is little predictive value between the images and that the two species being imaged do not have a clear correlation [44].

### 4.11. Flow Cytometry

After exposure, cells were washed three times with PBS. To detach J774A.1 cells, 1 mL of Accutase (Cat. #00-4555-56, Thermo Fisher Scientific) was applied for 5–10 min and incubated at 37 °C. In addition, a cell scraper (Sarstedt, Sevelen, Switzerland) was used and J774A.1 cells were collected in a flow cytometry tube (5 mL Polystyrene Round-Bottom Tube, Corning^®^ Falcon, Reinach, Switzerland). Caco-2 cells were detached with 0.05% Trypsin-EDTA in PBS (*v*/*v*), applied on the apical (0.3 mL) and basolateral side (1.5 mL) for 5 min at 37 °C, followed by the addition of 5 mL of cRPMI. Cells were centrifuged for 5 min at 500× *g*, washed in PBS, and then re-suspended in flow cytometry buffer, containing 1% BSA and 1 mM EDTA (Sigma- Aldrich, Buchs, Switzerland) in PBS at pH 7.0–7.4. Cells were stained with DAPI at the final concentration of 0.5 µg/mL, diluted in flow cytometry buffer (*v*/*v*) for 10 min at 4 °C. Data acquisition was performed on a BD LSR FORTESSA (BD Biosciences, San Jose, CA, USA) equipped with a violet laser (λ_ex_: 405 nm; DAPI), a blue laser (λ_ex_: 488 nm; SiO_2_-BDP FL NPs), and emission bandpass filters 450/50 (DAPI) and 530/30 (SiO_2_-BDP FL NPs). Flow cytometry data collected from 20.000 events were analyzed using the FlowJo software (Version 10.8.1, TreeStar, Woodburn, OR, USA).

### 4.12. Statistical Analysis

Statistical analysis was performed using GraphPad Prism 8 software (GraphPad Software Inc., San Diego, CA, USA). One-way analysis of variance (ANOVA; α = 0.05) with Dunnett’s or Tukey’s multiple comparisons tests was used to compare values among the different treatments. Statistically significant values among the treatments are shown in the figure captions.

## Figures and Tables

**Figure 1 toxins-15-00482-f001:**
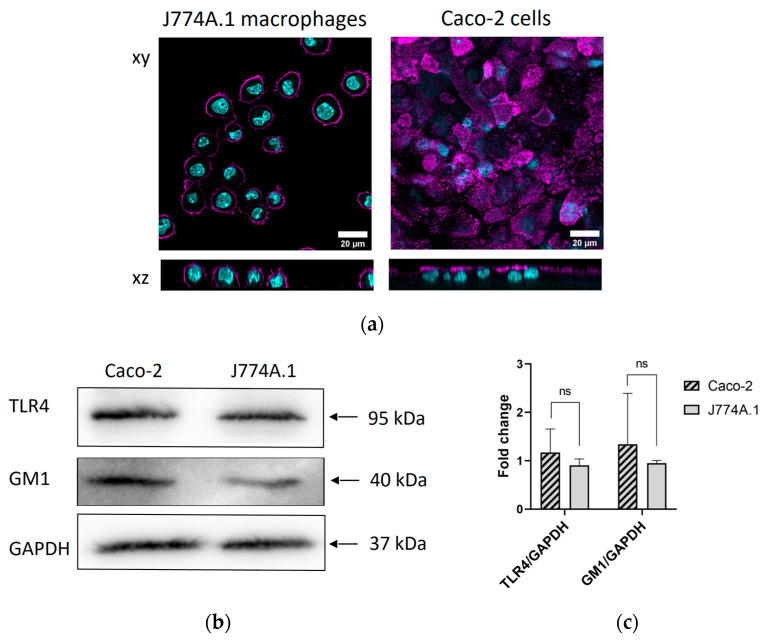
Cell surface expression of GM1 and LPS receptors. (**a**) Expression of ganglioside receptor (GM1)—a high-affinity binding site for cholera toxin subunit B (CTxB) in J774A.1 macrophages and differentiated Caco-2 cells as determined via confocal laser scanning microscopy. GM1 is shown in magenta and cell nuclei in cyan. Scale bar: 20 µm. (**b**) The expression of Toll-like receptor 4 (TLR4; 95 kDa)—the key receptor involved in LPS recognition and GM1 in J774A.1 and Caco-2 cells was analyzed via Western blot. Glyceraldehyde-3-phosphate dehydrogenase (GAPDH; 37 kDa) was used as an internal control for protein loading. (**c**) The average expression values of the indicated proteins upon normalization against the loading control GAPDH. The data are presented as mean ± standard error of mean (n = 3). Statistically significant differences among the groups were calculated using the Student’s *t*-test: ns-not significant. Full Western blot images including the corresponding controls and weight marker are shown in Appendix A.

**Figure 2 toxins-15-00482-f002:**
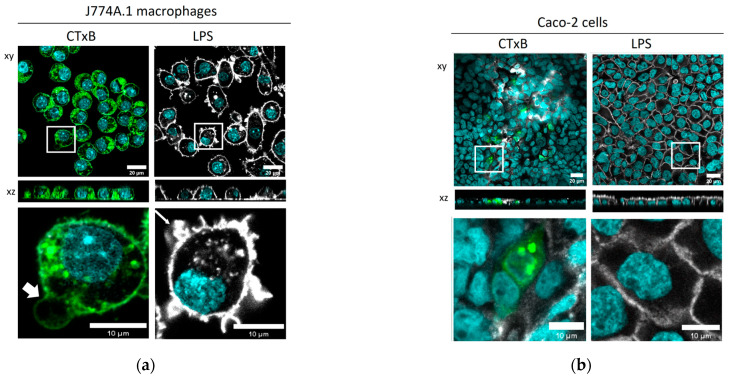
The effect of CTxB and LPS on cells’ morphology. Confocal laser scanning microscopy (CLSM) with zoomed-in images of single cells (**lower panel**) revealing the interaction of CTxB and LPS with (**a**) J774A.1 macrophages and (**b**) intestinal epithelial cells Caco-2 after 24 h of exposure to 1 µg/mL CTxB or 1 µg/mL LPS. Thicker and thinner white arrows indicate the membrane deformation and formation of membrane ruffles of J774A.1 macrophages, caused by CTxB and LPS, respectively. Cell nuclei (cyan), F-actin cytoskeleton (grey), and CTxB (green). Scale bar: 20 μm. Zoom-in images of the insets are shown below each image, scale bar: 10 µm.

**Figure 3 toxins-15-00482-f003:**
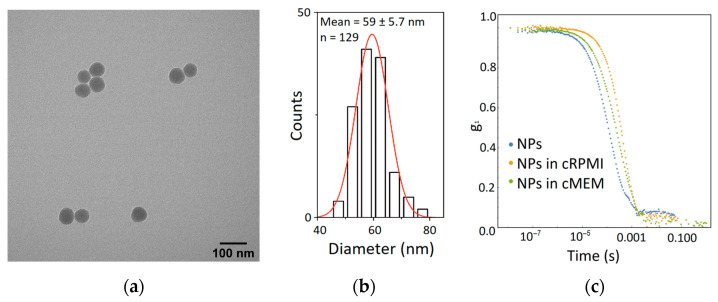
SiO_2_-Bodipy fluorescein (BDP FL) NP characterization. (**a**) Transmission electron micrograph of SiO_2_-BDP FL NPs (dTEM = 59 ± 6 nm). (**b**) Core diameter distribution of 59 nm SiO_2_-BDP FL NPs. Statistical analysis and graphical presentations were performed using the software Origin 2016 (OriginLab Corporation, Northampton, MA, USA). (**c**) The correlation functions of DLS measurements performed in Milli-Q water and supplemented medium at 24 h indicate stabile NP suspension. The data were obtained and analyzed by following a previously described method [30,31].

**Figure 4 toxins-15-00482-f004:**
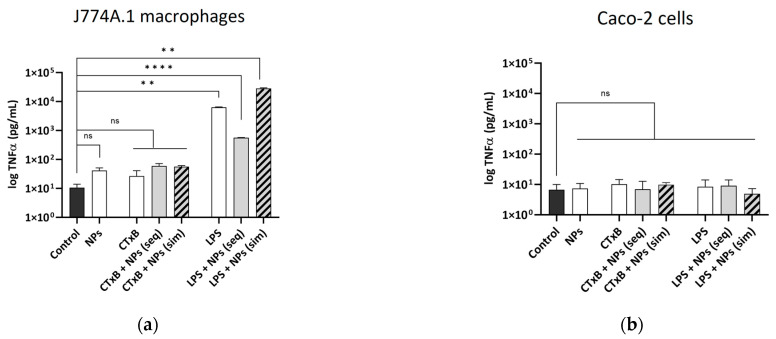
CTxB and LPS effect on TNF-α cytokine secretion. The secretion of pro-inflammatory mediator TNF-α by (**a**) J774A.1 cells and (**b**) Caco-2 cells after single, sequential (seq), and simultaneous (sim) co-exposure with NPs. The data for TNF-α secretion are obtained via ELISA and represented in pg/mL. The controls denote the mean values of the untreated cells. Data are presented as mean ± standard deviation (*n* = 3), analyzed via one-way ANOVA with Dunnett’s post hoc test for multiple comparisons; ns—not significant; ** *p* ≤ 0.01; **** *p* ≤ 0.0001.

**Figure 5 toxins-15-00482-f005:**
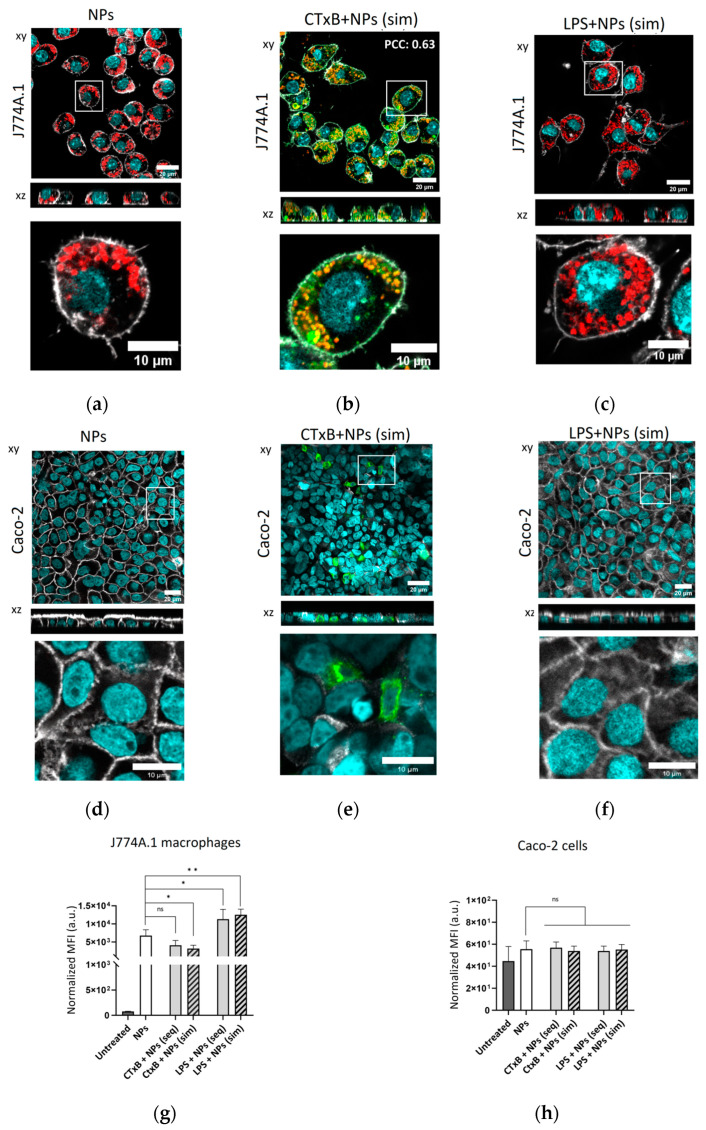
NP uptake upon cell exposure to CTxB and LPS. Confocal laser scanning micrographs with zoomed-in images of single cells (lower panel) of J774A.1 macrophages exposed to (**a**) single NPs or simultaneously to (**b**) NPs and CTxB-Alexa Fluor 647 or (**c**) NPs and LPS for 24 h, demonstrating high NP uptake. A colocalization between CTxB and NPs in J774A.1 macrophages is shown (yellow pixels) and was determined by the Pearson correlation coefficient (PCC) using Fiji software (version 2.9.0) with the JACoP plugin (*n* = 10 cells). Confocal laser scanning micrographs of Caco-2 cells showing no internalized NPs under (**d**) single NP exposure, (**e**) simultaneous NP co-exposure with CTxB-Alexa Fluor 647, or (**f**) simultaneous NP co-exposure with LPS for 24 h. Cell nuclei (cyan), cytoskeleton (grey), NPs (red), CTxB (green). Scale bar: 20 μm. Zoom-in images of the insets are shown below each image, scale bar: 10 µm. Median fluorescence intensity (MFI) of BDP FL signal, denoting cellular uptake of 59 nm SiO_2_-BDP FL NPs into (**g**) J774A.1 macrophages and (**h**) Caco-2 cells, assessed via flow cytometry. Upon J774A.1 macrophages’ exposure to CTxB, we observed a two-fold decrease in NP uptake, whereas simultaneous exposure of macrophages to LPS resulted in a two-fold increase in NP uptake. In Caco-2 cells, no uptake of NPs was observed across all exposure conditions. Data are expressed as mean + standard deviation of the mean (*n* = 3). Statistical significance was determined by one-way ANOVA with Dunnett’s post hoc test for multiple comparisons; * *p* ≤ 0.05; ** *p* ≤ 0.01; ns—not significant.

**Figure 6 toxins-15-00482-f006:**
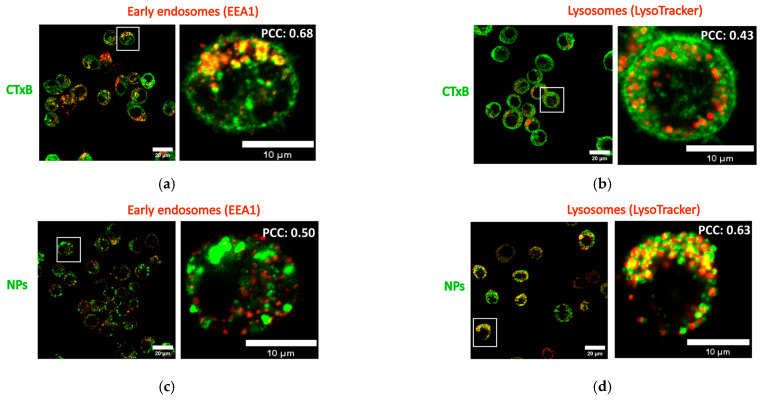
Intracellular localization of NPs in macrophages in the presence of CTxB. CTxB colocalization between CTxB and (**a**) early endosomes (EEA1) and (**b**) lysosomes (LysoTracker Red) of J774A.1 macrophages at 24 h incubation time; 59 nm SiO_2_-BDP FL NPs colocalization with (**c**) early endosomes (EEA1) and (**d**) lysosomes (LysoTracker Red). CTxB and NPs are shown separately in green and EEA1/LysoTracker staining in red. The colocalization (yellow pixels) was determined with Pearson’s correlation coefficient (PCC) using Fiji-based software with the JACoP plugin (n = 10 cells). PCC values are represented at the top right corner of the zoomed-in images.

**Table 1 toxins-15-00482-t001:** Nanoparticle size and size distribution were determined via DLS in water and complete cell culture media (cRPMI and cMEM) in the absence and presence of CTxB and LPS. All measurements were performed after 24 h incubation at 37 °C.

Solution	NPs	CTxB + NPs	LPS + NPs
Milli-Q water	85 ± 1 nm	83 ± 2 nm	82 ± 2 nm
cRPMI	95 ± 2 nm	96 ± 2 nm	98 ± 3 nm
cMEM	94 ± 2 nm	97 ± 2 nm	87 ± 1 nm

Data represent hydrodynamic diameter (nm) measured via DLS.

## Data Availability

The data presented in this study are available on request from the corresponding author.

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
