# Peer review of "The Functions of Cholera Toxin Subunit B as a Modulator of Silica Nanoparticle Endocytosis"

_toxins, 2023, doi:10.3390/toxins15080482_

Round 1
Reviewer 1 Report
The authors present an interesting study of silica nanoparticles and CTxB administered to cells. The experiments have been conducted thoroughly and the data is presented clearly. The biophysical size measurements help to reinforce that the differences in behaviour are not due to nanoparticle aggregation. I was not so surprised that TNF-alpha does not increase - I would have expected that the holotoxin would be required for that response. The co-localisation studies are convincing and the observation that CTxB reduces nanoparticle uptake could well be connected to its ability to change the curvature of the membrane as discussed earlier in the manuscript. Altogether I think this is a useful addition to the literature of CTxB and its effect on cellular delivery. - I suggest a couple of minor changes:
Line 27 “offer several benefits” – it would be helpful for the reader if one or two of the benefits could be listed in this sentence.
Line 135: the data in Figure 3 shows the nanoparticles are 59 +/- 6 nm, not +/- 2 nm as described in the text.
Author Response
Dear Reviewer 1,
We want to thank you for the effort and time dedicated to this manuscript. All the changes done in the manuscript are marked as track changes in the manuscript. Please see our responses below in red. We sincerely hope that these revisions adequately address your concerns.
The authors present an interesting study of silica nanoparticles and CTxB administered to cells. The experiments have been conducted thoroughly and the data is presented clearly. The biophysical size measurements help to reinforce that the differences in behaviour are not due to nanoparticle aggregation. I was not so surprised that TNF-alpha does not increase - I would have expected that the holotoxin would be required for that response. The co-localisation studies are convincing and the observation that CTxB reduces nanoparticle uptake could well be connected to its ability to change the curvature of the membrane as discussed earlier in the manuscript. Altogether I think this is a useful addition to the literature of CTxB and its effect on cellular delivery. - I suggest a couple of minor changes:
Line 27 “offer several benefits” – it would be helpful for the reader if one or two of the benefits could be listed in this sentence.
In the context of our study, the term "benefits" refers to the various applications of CtxB as drug and gene delivery carriers, as well as its use as stabilizing agents in therapeutics. Relevant references are added to these applications:
Lines 26-28: Among the variety of NPs used in biomedical applications, amorphous silica (SiO2) nanoparticles (NPs) are beneficial as drug and gene delivery carriers (10.3390/pharmaceutics12090826, 10.3390/pharmaceutics12070649) and as stabilizing agents in therapeutics (https://doi.org/10.1016/j.biopha.2022.113053).
Line 135: the data in Figure 3 shows the nanoparticles are 59 +/- 6 nm, not +/- 2 nm as described in the text.
The correction has been applied (59 +/- 6 nm).
Reviewer 2 Report
The article entitled “Functions of cholera toxin subunit B as a modulator of silica nanoparticle endocytosis” analyses variations in uptake behavior of 59nm SiO2 NPs in Caco-2 and J774A.1 cells in the presence of cholera toxin subunit B (and LPS) to mimic a possible scenario in the bacteria-rich gastrointestinal tract.
The experimental design is sound, and the conclusions are adequate considering the collected data; therefore, the article is suitable for publication on Toxins.
However, I suggest addressing a few points for the sake of clarity:
- First line of the introduction, SiO2: 2 needs subscript.
- Figure 1. I think it would be better also to show Western Blot for GM1. In both cases, (GM1 And TLR4) expression quantification normalized on the housekeeper in both cell lines would be beneficial.
- Figure 2. Why is F-actin staining not shown for cells treated with CTxB? The authors should include imaging for control cells in order to compare F-actin morphology.
- Since a dose-response analysis for CTxB and LPS is not reported, the authors should justify the choice of the concentration and incubation time.
- Figure 3. I think the panels should be larger with a more readable axis and legends.
- It would be important to show that after 5 days of dialysis, no more free dye is present in the SiO2 NPs and also show data to exclude dye leaching during in vitro experiments.
- Figure 5. For each confocal image, a zoomed inset would help the clarity. Again, why in the CTxB samples is there no F-actin staining? For panels g and h (flow cytometry) I think it would be better to include untreated samples to show the different fold increase of internalization when no toxin treatment is applied. It is not clear to me how the normalization is done. Usually, it is done on untreated samples. I don’t think this is the case, given the very high values of Fluorescent intensity…the author should clarify this. The graph in panel h could use a more expanded y-axis.
- Regarding the internalization pathways of CTxB, I wonder if the conjugation with AlexaFluor 647 might change the capability of CTxB to be recognized by the receptor GM1 (the fluorophore is about 10% of the total protein weight). Have the authors considered checking this possibility with some knock-out or competition experiment?
- Did the authors consider following CTxB co-localization with Golgi or ER? This would have strengthened their hypothesis.
The quality of English is acceptable
Author Response
Dear Reviewer 2,
Thank you for the dedicated effort and valuable time invested in evaluating this manuscript. All modifications made are indicated through the track changes in the manuscript. Please see the responses below in red. We sincerely hope that these revisions adequately address your concerns.
The article entitled “Functions of cholera toxin subunit B as a modulator of silica nanoparticle endocytosis” analyses variations in uptake behavior of 59nm SiO2 NPs in Caco-2 and J774A.1 cells in the presence of cholera toxin subunit B (and LPS) to mimic a possible scenario in the bacteria-rich gastrointestinal tract.
The experimental design is sound, and the conclusions are adequate considering the collected data; therefore, the article is suitable for publication on Toxins.
However, I suggest addressing a few points for the sake of clarity:
- First line of the introduction, SiO2: 2 needs subscript.
The correction has been applied.
- Figure 1. I think it would be better also to show Western Blot for GM1. In both cases, (GM1 And TLR4) expression quantification normalized on the housekeeper in both cell lines would be beneficial.
We included the western blot showing the expression of GM1 receptor in both cell lines. To provide comprehensive data, we have also included the expression levels of GM1 and TLR4, normalized to GAPDH, in both cell lines (Figure 1C).
The manuscript has been updated with the following modifications:
Lines 81-84: Additionally, we included the western blot data, showing the GM1 expression (Mw: 40 kDa) in both cell lines (Figure 1B). The GM1/GAPDH densitometry results showed no significant differences between Caco-2 cells and J774A.1 macrophages (Figure 1C).
Lines 87-89: The TLR4/GAPDH densitometry results showed no significant differences between Caco-2 cells and J774A.1 macrophages (Figure 1C).
Lines 101-104: The average expression values of the indicated proteins upon normalization against the loading control GAPDH. The data are presented as mean ± standard error of mean (n = 3). Statistically significant differences among the groups (Student t-test: ns-not significant).
Figure S2: Added figures (c) and (d), representing full western blot images with the corresponding weight marker lane showing expression of GM1 and GAPDH.
Additional experimental parameters for GM1 detection and densitometry have been introduced in the Materials and methods (section 4.4. Western blot).
- Figure 2. Why is F-actin staining not shown for cells treated with CTxB? The authors should include imaging for control cells in order to compare F-actin morphology.
F-actin staining was performed on all cells, including those treated with CTxB. However, in J774A.1 macrophages, the presence of CtxB on the membrane is overlapping with the F-actin staining. We have replaced Figure 2B with new images that display the F-actin staining. In Caco-2 cells, the cell borders can still be seen, but fainter and blurred. This finding suggests that CTxB may interfere with the labeling for F-actin.
The following sentence has been added:
Lines 121-123: In Caco-2 cells, fainter and blurred F-actin staining was observed, but the cell borders are still visible. This can be due to an interference of the fluorescently labelled CTxB with the F-actin label.
To compare the F-actin morphology, we have included confocal laser scanning microscopy (cLSM) images of control cells in Supplementary Figure S4.
- Since a dose-response analysis for CTxB and LPS is not reported, the authors should justify the choice of the concentration and incubation time.
The selection of LPS dose and incubation time was based on our previously published data (10.3390/cells9092099), wherein we demonstrated the J774A.1 macrophages response to the 1 µg/mL LPS. Our findings indicated that this concentration of LPS does not induce any cytotoxic effects, but does elicit membrane remodelling and pro-inflammaory response with detectable secretion of TNF-α at 24h of incubation. Additional in vitro studies on Caco-2 cells (10.3892/mmr.2021.11844, 10.1016/j.foodchem.2016.04.067) showed that 1 μg/ml of LPS modulates cells' morphology and instigates an inflammatory response without causing cell death.
Furthermore, we would like to direct the reviewer to two relevant references (10.1128/IAI.00581-06, 10.1016/j.molimm.2015.05.008), demonstrating the effects of 1 µg/mL CtxB on changing cellular responses in antigen presenting cells. These references provide further support for the rationale behind our chosen experimental conditions.
The following justification has been added to the manuscript:
Lines 128-136: The selection of LPS dose (1 µg/mL) and incubation time (24 h) was determined based on our previously published data, wherein we demonstrated the J774A.1 macrophage membrane remodeling and inflammatory response to the LPS [18]. Additional in vitro studies on Caco-2 cells demonstrated that 1 μg/ml of LPS modulates cells' morphology and instigates an inflammatory response without causing cell death (10.3892/mmr.2021.11844, 10.1016/j.foodchem.2016.04.067). For the choice of CtxB experimental conditions, we relied on the published findings of Schnitzer et al. [11] and Phongsisay et al. [12]. These studies provided valuable insights into the optimal conditions for studying the effects of CtxB on cellular responses.
- Figure 3. I think the panels should be larger with a more readable axis and legends.
We have increased the panel size and the axis/legends fond.
- It would be important to show that after 5 days of dialysis, no more free dye is present in the SiO2 NPs and also show data to exclude dye leaching during in vitro experiments.
We acknowledge the importance of dye leaching experiments in the context of NPs uptake studies, as suggested by the reviewer. Therefore, we conducted additional experiments specifically aimed at investigating the potential release of dye from SiO2-BDP FL NPs.
The data with the corresponding caption has been added to the Supplementary Figure S5.
Lines 167-168: By using fluorimetry, we demonstrated that the NPs remained stable, with no detectable dye leaching was observed (Figure S5).
For the expeirmental steps, please refer to the main manuscript – Materials and methods (Lines 341-353):
4.2. Dye leaching from SiO2 NPs
The potential release of dye from SiO2-BDP FL NPs after dialysis was assessed via fluorimetry. The NPs were subjected to high-speed centrifugation (16,000 × g) for 1 h, and the supernatants were collected. To ensure a minimal number of NPs remained in suspension, the supernatants were subsequently centrifuged again at the same speed. As a control, NPs in Milli-Q water at the administered dose (20 µg/mL) were included in the experiments. Fluorescence emission intensity measurements were performed using a Fluorolog TCSPC spectrofluorometer (Horiba, Northampton, UK) equipped with the FluorEssence software (v3.8). For each sample, an emission spectrum ranging from 500 to 600 nm with a fixed excitation wavelength (λex) of 488 nm was recorded. The excitation and emission slits were both set to a fixed width of 4 nm, ensuring consistent measurement conditions across all samples.
- Figure 5. For each confocal image, a zoomed inset would help the clarity. Again, why in the CTxB samples is there no F-actin staining? For panels g and h (flow cytometry) I think it would be better to include untreated samples to show the different fold increase of internalization when no toxin treatment is applied. It is not clear to me how the normalization is done. Usually, it is done on untreated samples. I don’t think this is the case, given the very high values of Fluorescent intensity…the author should clarify this. The graph in panel h could use a more expanded y-axis.
We have replaced Figure 5 with new images that display the F-actin staining and included the zoomed insets.
In Figures 5G and 5H where we show the flow cytometry analysis, we incorporated MFI od the fluorescent signal in untreated samples (cells stained with a viability dye, without any LPS or CTxB treatment). In addition, we expanded the y-axis of the graph in Figure 5H.
- Regarding the internalization pathways of CTxB, I wonder if the conjugation with AlexaFluor 647 might change the capability of CTxB to be recognized by the receptor GM1 (the fluorophore is about 10% of the total protein weight). Have the authors considered checking this possibility with some knock-out or competition experiment?
We appreciate the reviewer for raising this important question. Although we did not conduct knock-out or competition experiments in this study, we performed a literature research to investigate the potential effects of fluorophore conjugation on CtxB's binding capability to the GM1 receptor.
In support of our investigation, we referred to stoichiometry studies conducted by Maarouf Kabbani et al. (doi: 10.1073/pnas.2001119117), who demonstrated that the introduction of fluorescently labeled CTxB to GM1-containing membranes induced receptor clustering and the growth of membrane buds. Day et al. (doi: 10.1042/bse0570135) reported that fluorescently-labeled CTxB is capable of binding to multiple copies of the GM1 receptor and possesses the ability to locally remodel the membrane. Moreover, Rissanen et al. (doi: 10.3389/fphys.2017.00252) demonstrated that fluorescently labeled CTxB is a well-established tool for monitoring membrane domains in cell membranes. In another relevant study by Navolotskaya et al. (https://doi.org/10.1016/j.tiv.2017.12.010), 25I-labeled cholera toxin B subunit was found to bind to GM1 receptor of human Caco-2 intestinal epithelial cells with high affinity. These studies collectively suggest that the fluorophore conjugation does not significantly impede the binding capability of CTxB to the GM1 receptor.
- Did the authors consider following CTxB co-localization with Golgi or ER? This would have strengthened their hypothesis.
In our manuscript, we focused on evaluating the colocalization of CtxB and NPs with lysosomes and early endosomes. However, it is well-documented in the literature that CTxB can also undergo further accumulation in the Golgi or ER. While investigating the trafficking of CTxB to these intracellular organelles would indeed be an interesting avenue for future research, we determined that it was beyond the scope of our study. It has been previosly reported that the majority of the NPs accumulate in the intracellular vesicles, such as early endosomes and lysosomes and to a much lesser extend in the Golgi or ER (10.1002/smll.201000528, 10.1186/s12951-022-01670-9). Therefore, investigating the trafficking of CTxB to these organelles in the context of NPs would introduce a different focus of this study.
Round 2
Reviewer 2 Report
The authors improved the manuscript addressing all the raised concerns.